# Correction of Radiometry Data for Temperature Effect on Dark Current, with Application to Radiometers on Profiling Floats

**DOI:** 10.3390/s22186771

**Published:** 2022-09-07

**Authors:** Terence O’Brien, Emmanuel Boss

**Affiliations:** 1Institute for the study of Earth, Ocean and Space, University of New Hampshire, Durham, NH 03824, USA; 2School of Marine Sciences, University of Maine, Orono, ME 04469, USA

**Keywords:** radiometry, Argo floats, dark corrections

## Abstract

Measurements of daytime radiometry in the ocean are necessary to constrain processes such as photosynthesis, photo-chemistry and radiative heating. Profiles of downwelling irradiance provide a means to compute the concentration of a variety of in-water constituents. However, radiometers record a non-negligible signal when no light is available, and this signal is temperature dependent (called the dark current). Here, we devise and evaluate two consistent methods for correction of BGC-Argo radiometry measurements for dark current: one based on measurements during the day, the other based on night measurements. A daytime data correction is needed because some floats never measure at night. The corrections are based on modeling the temperature of the radiometer and show an average bias in the measured value of nearly 0.01 W m${}^{-2}$ nm${}^{-1}$, 3 orders of magnitude larger than the reported uncertainty of $2.5\;\times \;10^{-5}$ W m${}^{-2}$ nm${}^{-1}$ for the sensors deployed on BGC-Argo floats (SeaBird scientific OCR504 radiometers). The methods are designed to be simple and robust, requiring pressure, temperature and irradiance data. The correction based on nighttime profiles is recommended as the primary method as it captures dark measurements with the largest dynamic range of temperature. Surprisingly, more than 28% of daytime profiles (130,674 in total) were found to record significant downwelling irradiance at 240–250 dbar. The correction is shown to be small relative to near-surface radiance and thus most useful for studies investigating light fields in the twilight zone and the impacts of radiance on deep organisms. Based on these findings, we recommend that BGC-Argo floats profile occasionally at night and to depths greater than 250 dbar. We provide codes to perform the dark corrections.

## 1. Introduction

Sunlight fuels primary production in the oceans through microbial photosynthesis and is the primary source of thermal energy to the upper ocean. Accurate estimates of global primary production, oceanic photo-oxidation and thermal transfer are essential for quantifying both ocean carbon capture and long-term carbon storage in the deep ocean, as well as for providing radiative forcing for oceanographic and meteorological models. Downwelling planar irradiance, $E_{d}$, throughout the water column is one of the fundamental optical measurements from which the diffuse attenuation coefficient, $K_{d}$ [m${}^{-1}$], an apparent optical property, is derived. Additionally, vertical profiles of the spectral diffuse attenuation allow important water constituents such as chlorophyll and colored dissolved organic concentrations to be estimated [1,2].

The Argo program is a global array of profiling floats funded by national agencies. Since its first deployment in 1999, the array of Argo floats has grown to nearly 4000. These profile from the surface to 2000 dbar every 10 days, collecting CTD data. The project has expanded into Biogeochemical (BGC)-Argo by including optical, oxygen, nitrate and pH sensors on some floats [3]. Because the floats experience a dramatic range of temperature and pressure, and the sensors are not calibrated after deployment, it is essential to investigate the dynamics of sensor behavior. Without the retrieval of the sensors post-deployment, this must be done through investigation of the collected data.

Radiometers report a non-negligible output, known as the ’dark current’, even in the complete absence of ambient light. Furthermore, this dark current is known to display a temperature dependence. This is the reason why some commercial radiometers (e.g., SeaBird’s Hyper-OCR) have shutters allowing dark measurements to be taken in between readings of ambient light. SeaBird’s OCR504 radiometers, however, which are installed on the majority of BGC-Argo floats, do not have shutters (shutters increase energy consumption and cost). These radiometers have been shown to have a temperature-dependent dark response up to 2 or more times the known sensitivity of $2.5\;\times \;10^{-5}$ W m${}^{-2}$ nm${}^{-1}$ for $E_{d}$ (380 nm, 412 nm, 490 nm) [4,5]. These sensors have an additional channel measuring the intensity of photosynthetically available radiation (PAR), which has also been found to exhibit a temperature-dependent dark response [4,5]. To accurately characterize oceanographic processes at depth or in low-light conditions, where uncertainties in the radiometric measurements may be significantly impacted by uncertainties in the blank, a correct calibration which includes a correction for the temperature-sensitive dark current is essential [4,6]. Here, we investigate the dependence of the dark measurements (where measured irradiance is expected to be zero) on sensor temperature $T_{s}$ for radiometers on BGC-Argo floats and provide a quality control (QC) framework for correcting radiometer dark measurements for the instrument temperature dependence $dE_{dark}/dT_{s}$, [W m${}^{-2}$ nm${}^{-1}$ ${}^{\circ }$C${}^{-1}$] and $dPAR_{dark}/dT_{s}$, (μmol photons m${}^{-2}$ s${}^{-1}$ ${}^{\circ }$C${}^{-1}$) so that it can directly be applied by users. The analysis is done with data collected on floats characterized for this effect, which, as we show here, varies between individual radiometers in both magnitude and sign. We note that another paper with the same aims has been recently published, to which we have contributed [5]. However, the methods presented here are different and are intended to be applied directly to BGC-Argo s-files, rather than additionally using the Argo B- and transmission files (these contain data at float park depth, which we do not use here). Furthermore, unlike [5], we found no significant sensor drift over the lifetime of the floats analyzed once the temperature-dependent correction was applied.

## 2. Materials and Methods

Data from 218 BGC-Argo floats equipped with OCR504 radiometers, downloaded from https://www.ifremer.fr/erddap/tabledap/ArgoFloats.html (accessed on 28 March 2022), were investigated in this study. $E_{d}$ at three wavelengths, 380 nm, 412 nm, and 490 nm (W m${}^{-2}$ nm${}^{-1}$) and the instantaneous photosynthetically available radiation (iPAR, μmol photons m ${}^{-2}$ s${}^{-1}$ from 400–700 nm) were used. The floats were located across the global ocean, samplng a range of conditions, from continental shelves to open ocean gyres and from high to low latitudes. Radiometers may sample every 10 meters from 1000 m to 250 m, though many record no radiometric measurements at all in this interval. Starting at 250 m, radiometric measurements are made every 1 m, and from 10 m to the surface every 0.2 m. The average number of profiles taken per float in this dataset was 200. The average number of “good” daytime radiometry profiles per float was ninety, as determined following QC procedures outlined in [7], namely taken during consistent wave and cloud conditions and with sun elevation above 15° to the horizon. The average number of nighttime profiles taken per float, defined as sun elevations below the horizon, was six. The average temperature range experienced by these floats over their lifetime during good radiometric profiles in this dataset was 12.44 ${}^{\circ }$C.

With this dynamic temperature range and given that [4] showed the existence of a significant temperature response for these instruments, a temperature-dependent radiometric dark correction is necessary to accurately quantify or model processes occurring at low light levels. Sensor response to temperature varies between wavelengths for the same sensor and between sensors of the same model and may be positive or negative [4]. The response is dependent on sensor temperature rather than the ambient temperature (as expected for a temperature effect on the sensor electronics). We initially investigated the response based on ambient temperature, but found this inadequate as it exhibited a hysteresis, especially in regions with a pronounced thermocline. For this reason, a model of sensor temperature was developed similar to the one employed by [4].

Three approaches for quantifying temperature-dependent corrections for irradiance are investigated. Except where noted, the methods were identical for $E_{d}$ and PAR. The first two involved calculating a robust least squares regression on dark values (where irradiance is evaluated to be zero within the noise of the instrument) for sensor temperature ($T_{s}$) vs. measured irradiance ($E_{d}$). This provided a linear equation for the dark values of the form:(1)$$E_{d}\left(dark,T_{s}\right)\;=\;x_{0}+x_{1}\;\times \;T_{s},$$
where $x_{1}\;=\;dE_{d}/dT_{s}$ and $x_{0}$ is a constant (equivalent to $E_{d}\left(dark,T_{s}\;=\;0\right)$). The first method investigates night profiles, and the second investigates daytime profiles. The third method is designed to model the daytime profiles with a depth-dependent exponential + temperature dependent 1st degree polynomial. This further extended the range of depths where we could attempt to solve for the temperature sensitivity directly from daytime profiles. The model is:(2)$$E_{d}\left(z,T_{s}\right)\;=\;x_{0}+x_{1}\;\times \;T_{s}+x_{2}\;\times \;\left\{exp\left(-x_{3}\;\times \;\left(z-max\left(z\right)\right)\right)-1\right\},$$
where $x_{0}$ is the predicted irradiance $E_{d}(dark,T_{s}\;=\;0$, *z* = max(*z*)), $x_{1}$ is $dE_{d}/dT_{s}$, $x_{2}$ is a constant multiplier, and $x_{3}$ is the constant exponent for the depth-dependent (*z* is depth, positive downward) attenuation of irradiance. The model is fitted by the Levenberg–Marquardt method. While Equation (Equation 2) produced reasonable fits, the coefficients $x_{0}$ and $x_{1}$ showed a large range between profiles of the same float and, on average, had magnitudes significantly larger than those produced by Equation (Equation 1); they are thus assumed to represent a worse description of the temperature response of the sensor. We therefore decided not to use this model further.

### Profile Extraction, Quality Control and Modeling

The QC procedures outlined in [7] were followed to flag BGC-Argo radiometry profiles with unreasonable measurements or profiles taken during inconsistent wave or cloud conditions. Night profiles were determined based on sun elevation being less than 0 degrees above horizon at the specified latitude, longitude and time (using the routine SolarAzEl.m [8]). The dark portion of daytime profiles, occurring at depths where no light is detected, were determined using a lilliefors test for normality outlined by [7].

To ensure that the “dark” profiles were not influenced by light, we deployed a test to distinguish sensor noise from low levels of irradiance (e.g., moon and star light) when the values of irradiance measured approached the uncertainty of the radiometer. At great depth, assuming the optical properties of the water are constant, we expect downwelling light to display monotonic exponential decay (thus a monotonic linear decay of log($E_{d}$) with increasing depth) compared to random noise associated with the sensor. A least-squares regression of the depth (pressure) versus the log of the measured irradiance values was calculated for each profile. Any profile with a slope <−0.01 ($log_{10}$ (W m${}^{-2}$ db${}^{-1}$)) and a Spearman’s ρ>0.5 (meaning the decrease is monotonic) is assumed to be measuring significant downwelling irradiance. Such a slope is indicative of a consistent decline in irradiance significantly larger than the reported sensor uncertainty (=2.5 × 10${}^{-5}$ W m${}^{-2}$ nm${}^{-1}$).

For nighttime profiles that extend from ∼250 dbar to the surface, the test was applied three times to account for low levels of moonlight or starlight: from 150 dbar-surface, 100 dbar-surface, and 50 dbar-surface. For daytime profiles, the test was applied once, as the “dark” section generally spans a range of 10 m (240–250 dbar). For the daytime profiles (130,674 in total), 28% of “dark” profiles failed this test and were excluded from further analysis. For nighttime profiles (6281 in total), 51% fail at one of the depths (likely taken during twilight hours or under moonlight), with that section of the profile (from surface to given depth) removed from the regression analysis.

Following profile extraction, a model of the temperature-sensitivity of the dark current for each sensor was produced to correct for the effect of sensor temperature on the measured irradiance ($dE_{d}/dT_{s}$). We modeled the inherent lag in the sensor temperature by adjusting to that of the surrounding water column with a differential equation describing the relationship of the sensor temperature ($T_{s}$, unknown) to that of the water ($T_{env}$, measured by the float CTD sensor). The model is a first-order differential equation:(3)$$dT_{s}/dt\;=\;-\left(T_{s}-T_{env}\right)/k$$
that has the explicit solution (see Appendix A):(4)$$T_{s}\left(t\right)=T_{env}\left(0\right)exp\left(\frac{-t}{k}\right)+exp\left(\frac{-t}{k}\right)\int_{t^{\prime }=0}^{t^{\prime }}\frac{exp\left(\frac{t^{\prime }}{k}\right)\;\times \;T_{env}\left(t^{\prime }\right)}{k}\;dt,$$
where *k* is a time-lag constant. The rate of float rising was assumed to be constant with a value of 0.1 dbar/s [9]. We used *k* = 200 s (based on [4] and after finding no improvement upon exploring other values).

$T_{s}\left(t=0\right)$, the initial condition of sensor temperature, was set to approximate the temperature of the sensor 20 m below t=0 (thus 200 s previously), by calculating the average rate of change in the measured temperature $\left(dT_{env}/dz\right)$ over the 20 m range 250–230 dbar and setting $T\left(t=0\right)=T_{env}$ (250 dbar) −$dT_{env}/dz\;\times$ 20 m. This offset assumes a consistent gradient in temperature from 270–250 dbar and better models the temperature lag throughout the whole profile. If this resulted in a T(t=0) warmer than $T_{env}$(250 dbar), we required that $T\left(t=0\right)=T_{env}$ (250 dbar). For profiles with measurements made below 250 dbar, where sampling frequency was inconsistent, T(t=0) was set to $T_{env}\left(t\;=\;0\right)$, and data were linearly interpolated to a 1 dbar grid before the sensor temperature computation, with output only from sample depths saved.

A minimum/maximum range filter was applied to the irradiance profiles to remove remaining outlying values such as single spikes on otherwise good profiles, which may have been missed by previous filters. We constrained measurements to the range $|E_{d}|<0.03$ W m${}^{-2}$ nm${}^{-1}$ and |PAR|<50 μmol photons m ${}^{-2}$ s${}^{-1}$. These values were based on the distribution of measured irradiances from night profiles at depths greater than 300 dbar. In our dataset, 28% of nighttime values and 8% of daytime values were removed by this filter.

Following these QC steps, all the accepted profiles of a specific float and wavelength were compiled into a sensor-specific temperature versus irradiance database to determine a float-specific, wavelength-specific, temperature-dependent dark correction, which is assumed to be invariant in time. That is, the dark-current and temperature sensitivity were assumed constant throughout the life of a float, as we observed no evidence to the contrary. Daytime deep profiles and nighttime profiles were kept separate. The compiled profiles were then further subjected to the following two tests:

(a) Temperature range test: the temperature range of the compiled dark profiles must be greater than 2.5 ${}^{\circ }$C. This test is important as the $dE_{d}/dT_{s}$ is small (−0.003 to 0.003 W m${}^{-2}$ nm${}^{-1}$${}^{\circ }$C${}^{-1}$) and hence not detectable relative to other environmental processes if the temperature gradient in a profile is too small. Overall, 25% of night and 58% of day fits failed this test.

(b) Correlation test: Spearman’s rank correlation coefficient (ρ) between irradiance and temperature must have an absolute value greater than 0.3. Spearman’s tests how monotonic the relationship between two variables is (perfectly monotonic results in |ρ=1|). This test determines if the signal of temperature is likely influencing the irradiance value. Too small a |ρ| indicates it is likely undetectable in the available data. 13% of night and 17% of day data fits failed this test.

If both tests were satisfied, a robust linear fit (matlab robustfit.m) was computed from the compiled profiles of the specific float of sensor temperature ($T_{s}$) versus irradiance ($E_{d}$ or PAR) to produce a float-specific, wavelength-specific dark offset correction (Equation (Equation 1)). A robust fit was used rather than a normal least-squares regression to reduce the weight of possible outliers in the profiles. Where one or both tests were not satisfied, the median $E_{d}\left(dark\right)$ of all floats was used for $x_{0}$ with $x_{1}\;=\;0$, (e.g., $dE_{d}/dT_{s}\;=\;0$).

To ensure that we were not over-correcting for the temperature effect, we applied a filter based on the median ($x_{1}$) +/− 1.5 × IQR($x_{1}$) across all models of the same wavelength to decrease outlying values of $dE_{d}/dT_{s}$. IQR is the interquartile range, the distance between the 25th and 75th percentiles. Because the sensors are all of similar make and model, we expected a bound on the maximum temperature dependence of the dark measurement. Corrections that fall outside of the upper and lower bounds of the median($x_{1}$) +/− 1.5 × IQR($x_{1}$) threshold had ($x_{1}$) set to the threshold bound (upper or lower). $x_{0}$ was adjusted so that $dE_{d}/dT_{s}$ ($x_{1}$) intersects the median value of $E_{d}$ (dark) for that float by specifying that $x_{0}$ = median($E_{d}$) −$x_{1}\times$ median ($E_{d}$). A total of 8% of night profile values for $x_{1}$ and 2% of day profile values for $x_{1}$ were adjusted by this filter.

## 3. Results

Night profiles (method 1) and day profiles (method 2) produce comparable results for both correction parameters, $x_{0}$ (constant, [W m${}^{-2}$ nm${}^{-1}$]) and $x_{1}$ ($dE_{d}/dT_{s}$, (W m${}^{-2}$ nm${}^{-1}$${}^{\circ }$C${}^{-1}$)) (Equation (Equation 1), Figure 1 and Figure 2). Method 1 is recommended as the primary correction as it samples from the larger temperature range (encompassing conditions encountered by floats during their full profiles), produces more non-zero $x_{1}$ values (Figure 1) and, on average, is a smaller correction (Table 1).

The median temperature range of compiled profiles by method 2 was 1.71 ${}^{\circ }$C, with a median pressure range of 16.60 dbar. For comparison, the median temperature range of compiled nighttime profiles was 11.71 ${}^{\circ }$C with a median pressure range of 250 dbar. The median temperature range experienced by a float over the lifetime in our data set was 12.44 ${}^{\circ }$C. Method 2 produces more total corrections than method 1 (758 vs. 634), but method 1 produces a greater abundance of non-zero $x_{1}$ ($dE_{d}/dT_{s}$) (194 vs. 395). To visualize the cases of non-zero $x_{1}$, we display them separately (Figure 2). $x_{1}$ ranges from $-3.4\;\times \;10^{-3}$ to $2.3\;\times \;10^{-3}$ (W m${}^{-2}$ nm${}^{-1}$ ${}^{\circ }$C${}^{-1}$) by method 2 compared to $-2.4\;\times \;10^{-3}$ to $1.2\;\times \;10^{-3}$ (W m${}^{-2}$ nm${}^{-1}$ ${}^{\circ }$C${}^{-1}$) by method 1 (Figure 1). Method 2’s $x_{1}$ also shows greater variance with a larger standard deviation and interquartile range (Table 1). As such, method 1 produces a greater relative and absolute abundance of nonzero $dE_{d}/dT_{s}$ and produces a more constrained $dE_{d}/dT_{s}$ range than method 2. Values of the constant $x_{0}$ show, in general, a symmetric distribution around zero and of similar magnitude for all wavelengths (Figure 3).

Comparing the retrievals of $x_{1}$ for all floats that produced a correction by both methods, we find differences (Figure 4). For λ = 380 nm, 38 floats produced non-zero corrections for both methods, with a slope from robust regression = 1.12 and a Spearman’s ρ = 0.75. At λ = 412 nm, 35 floats produced non-zero corrections for both methods, with a ρ = 0.90 and slope = 1.24, indicating an over-prediction at 412 nm by the day-time method. At λ = 490 nm, 31 floats produced both non-zero corrections, with a slope = 1.29 and ρ = 0.70. Combining all λ (*n* = 104) the slope is 1.15 and ρ = 0.79 (not shown). For PAR, there is a strong correlation between the methods with a slope = 1.03 and ρ = 0.85. Overall, this suggests for $E_{d}$ a 15% overestimation by the day method compared to the night method, with λ-specific differences resulting in the largest overestimation by the night method compared to day at λ = 490 nm. However, at all λ and PAR, a Kolmogorov–Smirnov (k-s) test between the non-zero $x_{1}$ produced by both methods for the null acceptance results indicates the distributions are not different at the 5% significance level. Regressing $x_{0}$ values for all $E_{d}$ (not shown) returns a slope = 1.04 with ρ=0.85.

The correction is applied to each profile of a float as follows:(5)$$E_{d}\left(corrected\right)\;=\;E_{d}\left(measured\right)-\left[x_{0}+x_{1}\times T_{s}\right].$$

Statistics on absolute size of corrections applied by methods 1 and 2 on good profiles at all λ (19,605,908 measurements corrected) highlight the smaller average correction with smaller variance by the night method compared to day, though the differences are small (Table 2, Figure 5). Both corrections provide similar and consistent results when applied to profile data (Figure 6, Table 3).

## 4. Discussion and Summary

For this BGC-Argo dataset, the mean absolute temperature corrections on $E_{d}$ using night and day profiles are 0.008 and 0.0093 [W m${}^{-2}$ nm${}^{-1}$] and maximum absolute corrections are 0.044 and 0.0614 [W m${}^{-2}$ nm${}^{-1}$], respectively (Table 2). These corrections are 2-3 orders of magnitude larger than the known sensitivity of the sensors ($2.5\times 10^{-5}$W m${}^{-2}$ nm${}^{-1}$), are consistent with what has been observed in the lab by [4], and hence are significant. The average correction is O (10%) of the 0.1% light level, while the maximum is O (40%) of that value.

We further investigated whether the corrections had a significant impact on the diffuse attenuation coefficient:(6)$$K_{d}=-\frac{1}{E_{d}\left(z\right)}\frac{dE_{d}}{dz},$$
for profiles corrected with both methods using a center difference scheme. While we observe differences (Table 4), they are small (on the order of 0.001 m${}^{-1}$).

Thus, the correction does not produce a significant impact on $K_{d}$ at depth. As the temperature at depths is relatively constant, the impact of its gradient on $K_{d}$ is small (<4%). The measured values of $K_{d}$ are consistent with expected values for very clear waters, though higher than observed in the very clear waters of the Sargasso Sea [10,11].

The temperature correction for $E_{d}$ is likely to prove most important in studies investigating light fields in the twilight zone and the impacts of radiance on deep organisms, such as [12]. Additionally, it will impact investigations into the minimum light level supporting phytoplankton growth and on the impact of night time illumination on biology. Organisms in these conditions are extremely sensitive to low ambient light. Understanding their reaction to light requires accurate measurements at low irradiance conditions.

The method used here follows the work of [4], who has demonstrated the temperature dependence of the dark current of Satlantic OCR504 radiometers in the laboratory. Additionally, ref. [5] published a method approaching the same goals as ours, i.e., to produce a temperature-dependent dark correction for BGC-Argo profiles. Though our methods differ, we find overall agreement with both [4,5]. Here we provide a simple and robust method that allow users to carry out their own corrections and is consistent with both [4,5]. In [5], the BGC-Argo B- and transmission files are used in addition to BGC-Argo s-files, while ours is based only on s-files. The B- and transmission files are used to investigate measurements made at float park depth, while the s-files contain compiled profiles for each float. Ref. [5] investigated 55 floats. They provide a model that includes a drift correction, which we found no significant evidence for over the lifetime of the floats we analyzed. We recognize that by not investigating the measurements made at parking depth (instead basing our conclusion off of measurements made at deep profiles, where measurements may occasionally be as deep as ∼900 m), we are not using the best possible data to quantify a drift over the lifetime. However, as shown in Figure S9 of [5], coefficients for the drift correction have a maximum on the order of $1\times 10^{-7}$$days^{-1}$. This is smaller than the uncertainty in coefficients for the model constant and $dE_{dark}/dT$ (our $x_{1}$), and over a 1000 day lifetime, it produces a maximum correction on the order of $1\;\times \;10^{-4}$. After the drift correction, they fit a linear model to provide a temperature correction analogous to our Equation (Equation 1). In [4], 7 radiometers were tested in the laboratory over a temperature range of 26 ${}^{\circ }$C. They employed several methods for modeling the dark response: linear (such as ours), exponential, and quadratic. They chose the linear model as the primary model and only employed the quadratic or exponential if the $R^{2}$ value was significantly better. Out of 28 channels (7 radiometers × 4 channels), 17/28 were fit with the linear model, 4 with the exponential model, and 3 with the quadratic model, and for 4, no model fit well (Table 2a–g in [4]). The dynamic temperature range of their experiment compared to our in situ data (where average temperature range of a float lifetime is 12 ${}^{\circ }$C) may explain the necessity for a quadratic fit compared to our data (e.g., Figure 8 in [4]). The values of our modeled coefficients $dE_{dark}/dT$ agree well with [4]. Both show maximums on the order of $2\;\times \;10^{-3}$, larger than [5], whose maximum $dE_{dark}/dT$ are on the order of $4\;\times \;10^{-5}$ (Figure S9 in [5]). At all wavelengths and PAR, $dE_{dark}/dT$ is centered near zero, slightly biased towards negative values (decreasing dark signal with increasing temperature), and assumes a general Gaussian form. For $dPAR_{dark}/dT$, ref. [4] produces the smallest values, on the order of $2\;\times \;10^{-2}$, while [5] shows maximums of $4\times 10^{-2}$, and we have values as high as $-4\times 10^{0}$. Note, however, that we find a significantly higher model constant for PAR (our $x_{0}$) than either [4] or [5], with maximums two orders of magnitude larger than [4] and one order of magnitude larger than [5]. Our investigation of the daytime profiles revealed these significant dark readings at depth, and our corrections for PAR are of the same order relative to surface values as our corrections for $E_{d}$: at 10 m, our average PAR correction is on the order of 0.001% of the 10 m measured PAR value, analogous to the average 10 m correction at all three wavelengths. While we find some significant differences between our method and [4,5], the end-user applicability, robust approach, consistency between day and night methods (as in Figure 4), consistency between size of corrections applied across all four wavebands, and number of floats investigated (219) provide evidence for the utility of the methods presented in this paper.

Based on the data presented here and elsewhere [4,5], we recommend that a correction for the temperature effect on the dark current be applied to all radiometry data on floats. When no nighttime profiles are available, a correction based on daytime measurements is better than no correction (as it is highly correlated with the nighttime correction, when both are available). However, it is best if sufficient nighttime profiles are available, as the correction made with them seems superior (more consistent between sensors and lower over all). This is sensible given the larger dynamic range in temperature that it is based on. Expanding profiles of radiance to greater depths is likely to also improve the correction.

## Figures and Tables

**Figure 1 sensors-22-06771-f001:**
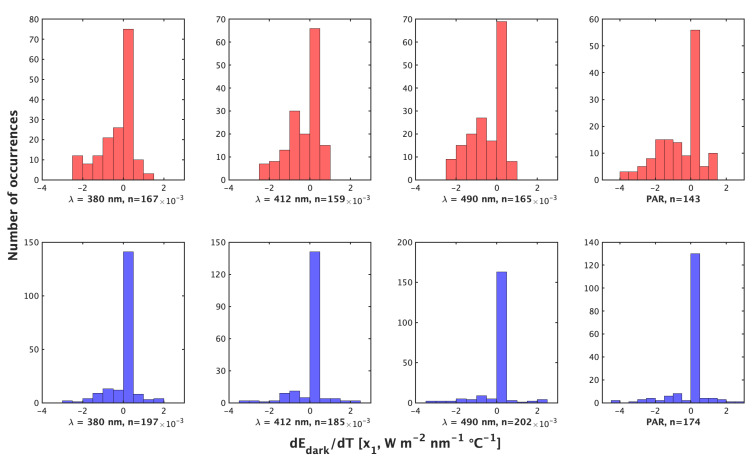
Histograms of the value of $x_{1}\;=\;dE_{d}/dT_{s}$ (W m${}^{-2}$ nm${}^{-1}$${}^{\circ }$C${}^{-1}$) by the night method (**top**, red) and Day method (**bottom**, blue) for λ = 380 nm, 412 nm, 490 nm, and iPAR (**left** to **right**).

**Figure 2 sensors-22-06771-f002:**
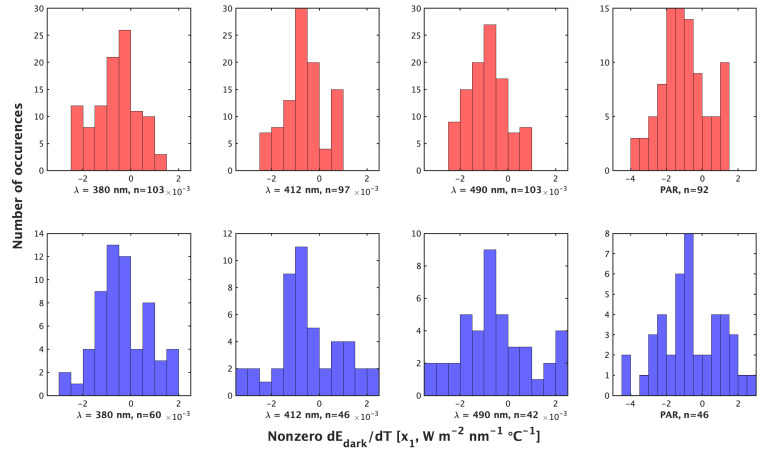
Histograms of the non-zero value of $x_{1}=dE_{d}/dT_{s}$ (W m${}^{-2}$ nm${}^{-1}$${}^{\circ }$C${}^{-1}$) or $dPAR/dT_{s}$ (μmol photons m${}^{-2}$ s${}^{-1}$${}^{\circ }$C${}^{-1}$) by the night method (**top**) and day method (**bottom**) for (**left** to **right**) λ = 380 nm, 412 nm, 490 nm, and PAR.

**Figure 3 sensors-22-06771-f003:**
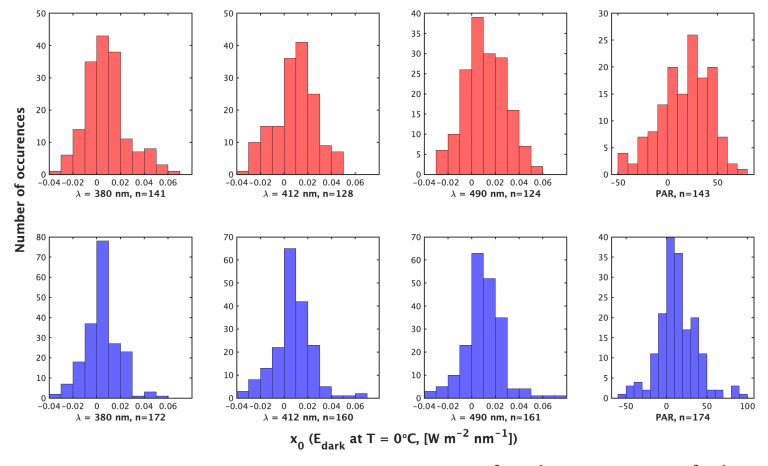
Histograms of the value of $x_{0}$ (W m${}^{-2}$ nm${}^{-1}$ or μmol photons m${}^{-2}$ s${}^{-1}$) by the night method (**top**) and day method (**bottom**) for (**left** to **right**) λ = 380 nm, 412 nm, 490 nm and PAR. $x_{0}$ is the value reported by the irradiance sensor in the dark at $T_{s}$ = 0 ${}^{\circ }$C.

**Figure 4 sensors-22-06771-f004:**
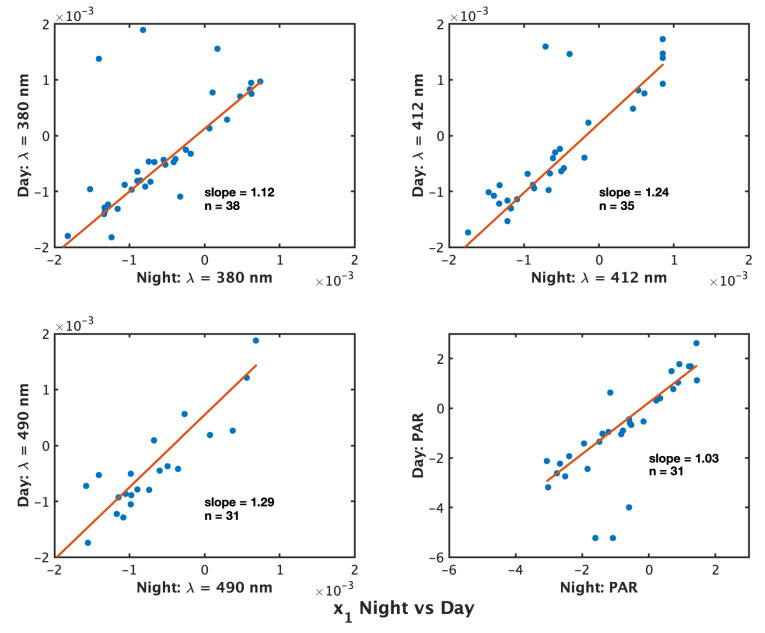
Comparison of $x_{1}$ by obtained from nighttime profiles (*x*-axis) and daytime profiles (*y*-axis) (W m${}^{-2}$ nm${}^{-1}$${}^{\circ }$C${}^{-1}$ or μ mol photons m${}^{-2}$ s${}^{-1}$${}^{\circ }$C${}^{-1}$) for floats that produced non-zero $x_{1}$ using both methods. Results for λ = 380 nm (**top left**), 412 nm (**top right**), 490 nm (**bottom left**), and PAR (**bottom right**).

**Figure 5 sensors-22-06771-f005:**
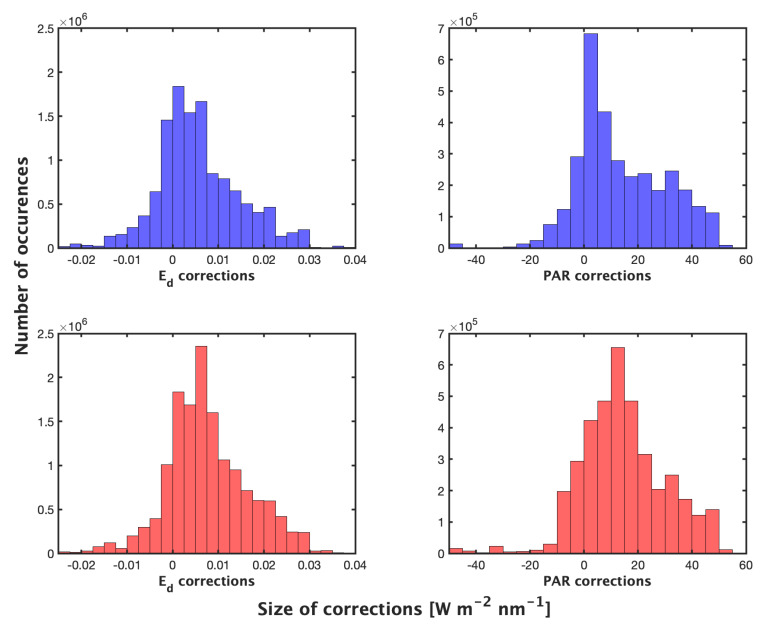
Size of corrections applied by the night method (**top**) and day (**bottom**) on good profiles at all wavelengths (19,605,908 measurements corrected) (W m${}^{-2}$ nm${}^{-1}$ and μmol photons m${}^{-2}$ s${}^{-1}$). Statistics shown in Table 2.

**Figure 6 sensors-22-06771-f006:**
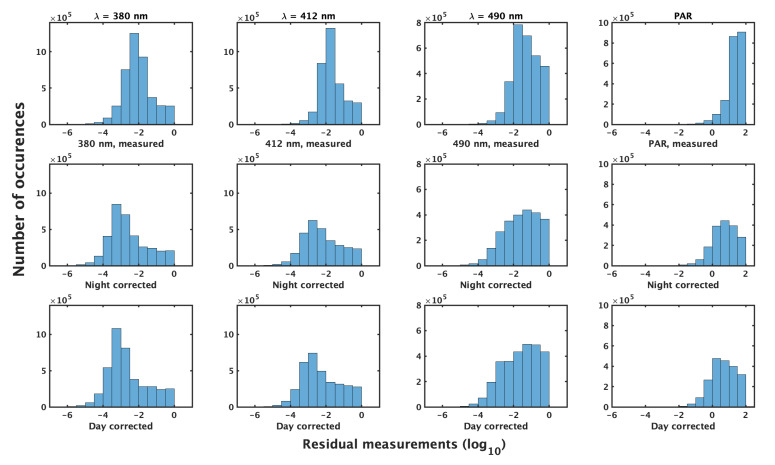
Measurements of $E_{d}$(λ, z) < 1 W m${}^{-2}$ nm${}^{-1}$ and PAR(z) < 100 μmol photons m${}^{-2}$ s${}^{-1}$ (**top** row) after corrections are applied by the night method (**middle** row) and day (**bottom** row). Columns are (**left** to **right**) λ = 380, 412, 490 nm and PAR. Plotted on $log_{10}$ scale.

**Table 1 sensors-22-06771-t001:** Nonzero $x_{1}$ by the night method and day method for all λ (W m${}^{-2}$ nm${}^{-1}$${}^{\circ }$C${}^{-1}$), as shown in Figure 2.

Method	Median	IQR	Mean	SD
Night $E_{d}$	−0.00067	0.001	−0.0007	0.00085
Day $E_{d}$	−0.00058	0.00173	−0.00044	0.0013
Night PAR	−1.064	1.674	−0.97	1.31
Day PAR	−0.953	2.858	−0.96	1.98

**Table 2 sensors-22-06771-t002:** Absolute size of corrections applied by both methods on good profiles at all wavelengths and PAR (19,605,908 measurements corrected) (W m${}^{-2}$ nm${}^{-1}$ or μ mol photons m${}^{-2}$ s${}^{-1}$).

Method	Max	Median	IQR	Mean	SD
Night $E_{d}$	0.0444	0.0057	0.0093	0.008	0.0072
Day $E_{d}$	0.0614	0.0071	0.0101	0.0093	0.0074
Night PAR	55.24	10.86	22.96	16.03	14.08
Day PAR	68.58	13.98	18	17.16	13.13

**Table 3 sensors-22-06771-t003:** Measurements for all λ of measured $E_{d}$(λ, z) < 1 W m${}^{-2}$ nm${}^{-1}$ and PAR(z) < 100 μmol photons m${}^{-2}$ s${}^{-1}$, corrected by the night and day methods (*n* = 12,930,490).

Method	Median	IQR	Mean	SD
Measured $E_{d}$	0.0166	0.0576	0.0905	0.1824
Night corrected $E_{d}$	0.0048	0.0512	0.082	0.1822
Day corrected $E_{d}$	0.0035	0.0505	0.082	0.1828
Measured PAR	26.53	32.37	31.78	24.79
Night corrected PAR	5.031	17.72	13.72	20.91
Day corrected PAR	3.90	17.77	13.19	20.85

**Table 4 sensors-22-06771-t004:** $K_{d}\left[m^{-1}\right]$ calculated on good daytime profiles, all λ, for measurements where 0.1 ≤ $E_{d}$(measured) ≤ 1 (W m${}^{-2}$ nm${}^{-1}$) (*n* = 1,741,267).

Method	Median	IQR	Mean	SD
Measured	0.052	0.013	0.053	0.035
Night corrected	0.051	0.013	0.053	0.034
Day corrected	0.05	0.014	0.051	0.031

## Data Availability

All data used in this study come from a public data base: https://erddap.ifremer.fr/erddap/tabledap/ArgoFloats.html (accessed on 28 March 2022). Scripts to perform the methods in Matlab are available at https://github.com/TOceans/ArgoRadiometryDark (accessed on 28 March 2022).

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
