# Peer review of "Correction of Radiometry Data for Temperature Effect on Dark Current, with Application to Radiometers on Profiling Floats"

_sensors, 2022, doi:10.3390/s22186771_

Round 1

Reviewer 1 Report

Measurements of radiometry in the ocean are critical. This paper devise and evaluate two consistent methods for post-life calibration and correction of BGC-Argo radiometry measurements which are important to compute the concentration of a variety of in-water constituents. The work  is of great significance for marine physics research. The  data are detailed and the experimental analysis is reasonable. But the current status of related work and the comparison with other methods are relatively weak.  

Author Response

Thank you for the thoughtful read-through and comments. Based on your statement that the comparison to related work is relatively weak, we have added a substantial paragraph in the discussion section, lines 266 - 309. In addition, we took your suggestion of a significant language and style revision and have edited language, including structure and tense, throughout the manuscript following an additional read by a native English speaker. 

Reviewer 2 Report

In this paper, the authors proposed a correction framework for the post-processing of the BGC-Argo radiometry measurements. This correction framework was temperature-dependent and was suggested to be applied to daytime and nighttime measurements separately. The results of such corrections were significant and necessary, which were two to three orders of magnitude larger than the sensitivity of the sensors. Generally, the manuscript is well organized and written. The method is feasible and the data analysis is clear and reliable. I would like to recommend this paper to be considered for publication on Sensors after minor revisions. Please find my comments below.

1. Can the authors briefly explain their initial motivation for doing the correction for both daytime and nighttime measurements?

2. The average number of gooddaytime profiles per float was ninety while nighttime profiles was only six. Why is there so little data in nighttime?

L54: BGC Argo -> BGC-Argo

L56: PAR -> Please give the full name of PAR.

L108: Quality-control (QC) -> The full name of QC has been given at L47.

L254: is likely to prove most important is in studies... -> remove is

Author Response

Question 1: Can the authors briefly explain their initial motivation for doing the correction for both daytime and nighttime measurements?

Question 2:The average number of “good” daytime profiles per float was ninety while nighttime profiles was only six. Why is there so little data in nighttime?

Response: Thank you for the careful read-through and helpful feedback. We have updated the manuscript per your comments. Please find answers to your questions below:

  1. The primary profiling mode for the BGC-Argo floats is to profile from depth to the surface every 10 days at local noon. Many floats may take occasional night profiles as well. The nighttime profiles cover a larger range of temperature and pressure and are therefore preferred for the computation of a temperature correction. We find an average of 90 daytime profiles and 6 nighttime profiles for floats. For floats without daytime profiles or where these prove inadequate, the day method is shown to be consistent.  We have added words into the abstract and text to make this point clearer to the readers.
  2. The low number of nighttime profiles compared to day is due to the primary profiling mode of the floats, which are designed to profile at local noon to capture daytime processes, such as irradiance profiles.

Minor comments:

L54: BGC Argo -> BGC-Argo

Thank you. Done as suggested.

L56: PAR -> Please give the full name of PAR.

Thank you. Done as suggested.

L108: Quality-control (QC) -> The full name of QC has been given at L47.

Thank you. Done as suggested. 

L254: is likely to prove most important is in studies... -> remove “is”

Thank you. Done as suggested.

Reviewer 3 Report

Line 43-45 : In regard to the term uncertainty you should try to explain why there is such a difference between the manufacturer specification and the outcome of your analysis. It may be that the manufacturer has a different understanding of the term. If resolution and uncertainty are treated as synonymous that would definitely be wrong.

- Line 93-95: Please, provide further details on why you have selected the mentioned functional dependence. If exponential functions are chosen then typically there is some physical reasoning behind.

- Line 149-155. I do not understand how you have selected the starting temperature. I would assume that there is always a small difference between sensor temperature and ambient water temperature, as the electronics will radiate some heat into the pressure housing. Do you agree ?

- Line 182: Are you planning to make the used correction algorithm as free exchange file available, for instance via GitHub?  

- Line 236ff: Besides the amount of correction could you also provide a number describing the uncertainty of the measurement results, for instance for Ed in figure 6? Maybe something like a standard deviations of the shown data could serve as an uncertainty estimate.

Author Response

Thank you for the careful read-through and helpful comments. Please find answers to your questions below:

Comment 1. Line 43-45 : In regard to the term uncertainty you should try to explain why there is such a difference between the manufacturer specification and the outcome of your analysis. It may be that the manufacturer has a different understanding of the term. If resolution and uncertainty are treated as synonymous that would definitely be wrong.

Response 1: Thank you. The manufacturer does not warn users that there is a temperature sensitivity to the dark current.The manufacturer does provide an uncertainty to the dark which is based on the standard deviation of the dark signal when there is no light impinging on the sensor.

Comment 2. Line 93-95: Please, provide further details on why you have selected the mentioned functional dependence. If exponential functions are chosen then typically there is some physical reasoning behind.

Response 2: Thank you. The model (equation 2) was an attempt at solving for the temperature dependence (dE(dark)/dT) from full radiometry profiles, rather than using only the dark portions of profiles as in equation 1. As such, the exponential term is included to account for the exponential decay of light based on the Lambert-Beer law. 

Comment 3. Line 149-155. I do not understand how you have selected the starting temperature. I would assume that there is always a small difference between sensor temperature and ambient water temperature, as the electronics will radiate some heat into the pressure housing. Do you agree ?

Response 3: Thank you. We agree that there is always a small difference between sensor temperature and ambient water temperature. During ascending profiles we assume that gradients in temperature are larger than temperature differences caused by the heating from the sensor’s internal electronics. Floats at park depth generally remain there for 10 days with electronics turned off to conserve battery; as such, at z = z(parking depth) we assume that the sensor temperature is equal to the ambient temperature. Radiometry profiles typically begin at z = 250 dbar, and as such we have employed the method described in the text to approximate the sensor temperature T(z = 250 dbar). Some irradiance profiles measure deeper than 250 db, which we include in our analysis; at these depths measurements are made irregularly with respect to depth, which introduces additional uncertainty into our approximation of T(z = zmax). At great depths we assume the temperature profiles are more likely to remain relatively constant in value.Therefore, for profiles measuring at greater depths, we use the initial ambient temperature as the starting temperature for our computation of the sensor temperature.

Comment 4. Line 182: Are you planning to make the used correction algorithm as free exchange file available, for instance via GitHub?  

Response 4: Yes.We have included a link to GitHub in our Data Availability section. 

Comment 5. Line 236ff: Besides the amount of correction could you also provide a number describing the uncertainty of the measurement results, for instance for Ed in figure 6? Maybe something like a standard deviations of the shown data could serve as an uncertainty estimate.

Response 5: Yes. The statistics associated with figure 6 are provided in Table 2, providing the Median, IQR, Mean, Standard Deviation, and Maximum corrections by each method.